# The SAM domain of mouse SAMHD1 is critical for its activation and regulation

Olga Buzovetsky[1], Chenxiang Tang[1], Kirsten M. Knecht[1], Jenna M. Antonucci[2], Li Wu [2],
Xiaoyun Ji[3] & Yong Xiong[1]

Human SAMHD1 (hSAMHD1) is a retroviral restriction factor that blocks HIV-1 infection by depleting the cellular nucleotides required for viral reverse transcription. SAMHD1 is allosterically activated by nucleotides that induce assembly of the active tetramer. Although the catalytic core of hSAMHD1 has been studied extensively, previous structures have not captured the regulatory SAM domain. Here we report the crystal structure of full-length SAMHD1 by capturing mouse SAMHD1 (mSAMHD1) structures in three different nucleotide bound states. Although mSAMHD1 and hSAMHD1 are highly similar in sequence and function, we find that mSAMHD1 possesses a more complex nucleotide-induced activation process, highlighting the regulatory role of the SAM domain. Our results provide insights into the regulation of SAMHD1 activity, thereby facilitating the improvement of HIV mouse models and the development of new therapies for certain cancers and autoimmune diseases.

[1] Department of Molecular Biophysics and Biochemistry, Yale University, New Haven, CT 06520, USA. [2] Center of Retrovirus Research, Department of Veterinary Biosciences, The Ohio State University, Columbus, OH 43210, USA. [3] The State Key Laboratory of Pharmaceutical Biotechnology, School of Life Sciences, Nanjing University, Nanjing, Jiangsu 210023, China. Olga Buzovetsky and Chenxiang Tang contributed equally to this work. Correspondence and requests for materials should be addressed to X.J. (email: xiaoyun.ji@nju.edu.cn) or to Y.X. (email: yong.xiong@yale.edu)

The sterile alpha-motif (SAM) and histidine-aspartate (HD) domain-containing protein 1 (SAMHD1) is a dNTP phosphohydrolase that restricts viral replication by limiting the cellular dNTP pool[1–7]. Without an adequate supply of dNTPs, retroviruses like HIV-1 cannot complete reverse transcription. In addition to its role in the antiviral response, SAMHD1 is also implicated in the autoimmune disease Aicardi–Goutieres syndrome (AGS). Homozygous mutations in the *SAMHD1* gene lead to the accumulation of nucleotides in the cell and result in symptoms that mimic a congenital viral infection[8,9]. This highlights the importance of SAMHD1 activity in the human immune system and dNTP metabolism[6,10,11].

Like hSAMHD1, mSAMHD1 restricts HIV-1 through its dNTPase activity, and the activities of both enzymes are tightly regulated by an allosteric activation mechanism[12–16]. The active tetramer form of SAMHD1 is induced by cellular nucleotides, which bind two allosteric sites of each subunit. While allosteric site (Allo-site) 1 accommodates only GTP or dGTP, Allo-site 2 permits any dNTP[12,15,17]. The nucleotide-induced assembly of the SAMHD1 tetramer is necessary for its antiviral restriction activity[15–17]. Interestingly, the HD domain of hSAMHD1 was found to have higher enzymatic and antiviral activities than the full-length enzyme[15], suggesting a potential regulatory role for the SAM domain. Without a structure of full-length SAMHD1, it has previously been difficult to determine the role of the SAM domain. In addition, despite past investigations on hSAMHD1, mechanisms for mSAMHD1 activation and viral restriction remain unclear.

Two isoforms of mSAMHD1 resulting from alternative splicing share about 72–74% sequence identities with hSAMHD1[18]. These isoforms are identical until residue 593 where the C-termini differ. While isoform 1 (iso1) has a phosphorylation site at residue T634 that regulates its antiviral activity, isoform 2 (iso2) lacks this site. Phosphorylation of the corresponding residue in hSAMHD1 (T592) has been shown to negatively regulate SAMHD1's ability to restrict HIV by affecting the formation of an active tetramer[19–24]. The effect of phosphorylation of mSAMHD1 iso1 is still unclear, although studies suggest that the T634D phosphomimetic mutation leads to loss of HIV-1 restriction in non-dividing cells, it has no effect on murine leukemia virus infection in dividing cells[18]. This suggests that different mechanisms of regulating retroviral restriction exist between hSAMHD1 and mSAMHD1.

In this study, we explore the structural and functional differences between human and mouse SAMHD1. While both mSAMHD1 and hSAMHD1 are allosterically activated by nucleotides, we find that the two enzymes have different assembly processes for the active tetramers. Furthermore, our results show that the role of the SAM domain differs between mSAMHD1 and hSAMHD1. The SAM domain is not required for dNTP hydrolysis and viral restriction by hSAMHD1, but is essential for these activities of mSAMHD1. To better understand the regulatory role of the SAM domain, we crystallized full-length mSAMHD1 in three different states (0, 1, and 2 allosteric nucleotides bound). In addition to capturing important interactions between the SAM and HD domains, these structures delineate the mSAMHD1 allosteric activation process that governs SAMHD1 enzymatic activities. These results are important for the assessment of SAMHD1 as a potential therapeutic target for HIV-1 infection and autoimmune diseases, such as AGS.

## Results

### SAM domain of mSAMHD1 is required for tetramerization.
The oligomerization states of purified mouse and human SAMHD1 are different in the absence of added nucleotides.

hSAMHD1 exists as a monomer in a nucleotide-free purification buffer, as monitored by both size-exclusion chromatography (SEC) and analytical ultracentrifugation sedimentation velocity (AUC). However, under the same conditions, mSAMHD1 is predominantly a dimer with some population of tetramers (Fig. 1a–c). After addition of the non-hydrolyzable nucleotide dGTP-α-S, hSAMHD1 readily forms tetramers, but a significant portion of mSAMHD1 remains as dimer (Fig. 1a–c). In fact, complete tetramer formation of mSAMHD1 requires up to 24 h of assembly with dGTP-α-S (Supplementary Fig. 1a), suggesting that mSAMHD1 has a more complex and slower process of tetramer assembly.

To determine whether the SAM domain plays a role in the tetramer formation of mSAMHD1, we repeated these assays with truncated proteins that do not contain the SAM domains, namely mouse SAMHD1 HD domain (mHD, residues 145–658 and 145–651 for iso1 and iso2, respectively). mHD exists in an equilibrium of monomers and dimers in the absence of allosteric nucleotides (Fig. 1f), similar to what has been observed with human SAMHD1 HD domain (hHD, residues 113–626)[12,15]. While only monomers of mHD were observed by SEC (Fig. 1e), a substantial amount of mHD dimers are detected by AUC (Fig. 1f). This may be due to sample dilution or the stripping of potential nucleotides co-purified with the mHD dimer during the SEC run. Importantly, this contrasts with the behavior of full-length mSAMHD1, which readily purifies as dimers in the absence of additional nucleotides (Fig. 1b, c). This suggests that in the absence of nucleotides the SAM domain is important for mSAMHD1 oligomerization. Surprisingly, upon addition of dGTP-α-S, mHD forms significantly less tetramers than hHD does (Fig. 1e, f). These experiments demonstrate that mHD is deficient in forming a stable tetramer, highlighting important differences in the functions of the SAM domains of mouse and human SAMHD1. Although full-length mSAMHD1 inherently tends to oligomerize in the absence of nucleotides (Fig. 1b), this tendency is modestly enhanced in the presence of nucleotides. In this regard, mSAMHD1 is less sensitive to nucleotides for oligomerization than hSAMHD1.

### HD domain of mSAMHD1 is insufficient for dNTPase activity.
Like hSAMHD1, mSAMHD1 is a dNTPase that reduces the dNTP level in mouse cells and restricts retroviral infection[25]. To confirm the catalytic activity of mSAMHD1, we analyzed nucleotide hydrolysis by both isoforms of the enzyme in vitro. We incubated highly purified recombinant mSAMHD1 protein with dGTP and monitored subsequent reaction products using high-performance liquid chromatography (HPLC) (Fig. 1g) and a malachite green (MG) colorimetric assay (Supplementary Fig. 1b). These two independent assays show that both isoforms of full-length mSAMHD1 and hSAMHD1 exhibit similar rates of dNTP hydrolysis (Fig. 1g, Supplementary Fig. 1b).

Surprisingly, our activity assay revealed that mHD is not sufficient for catalytic activity (Fig. 1g, Supplementary Fig. 1b). We found that although mHD displays residual dNTPase activity over time, the hydrolysis rate is drastically reduced from that of the full-length enzyme (Fig. 1h). This is in stark contrast to hHD, which has higher dNTPase activity than full-length hSAMHD1[2,12,26–28] (Fig. 1g). Since removing the SAM domain leads to opposite effects on the activities of human and mouse SAMHD1, the regulatory role of the SAM domain likely differs between the two enzymes. Consistently, mHD was also deficient in tetramer formation (Fig. 1e, f), suggesting that the SAM domain contributes to the allosteric activation of mSAMHD1.

We further assessed the HIV-1 restriction activity of mHD. While the full-length mSAMHD1 was able to restrict HIV-1 infection, mHD increased viral infection by 1.6-fold compared to

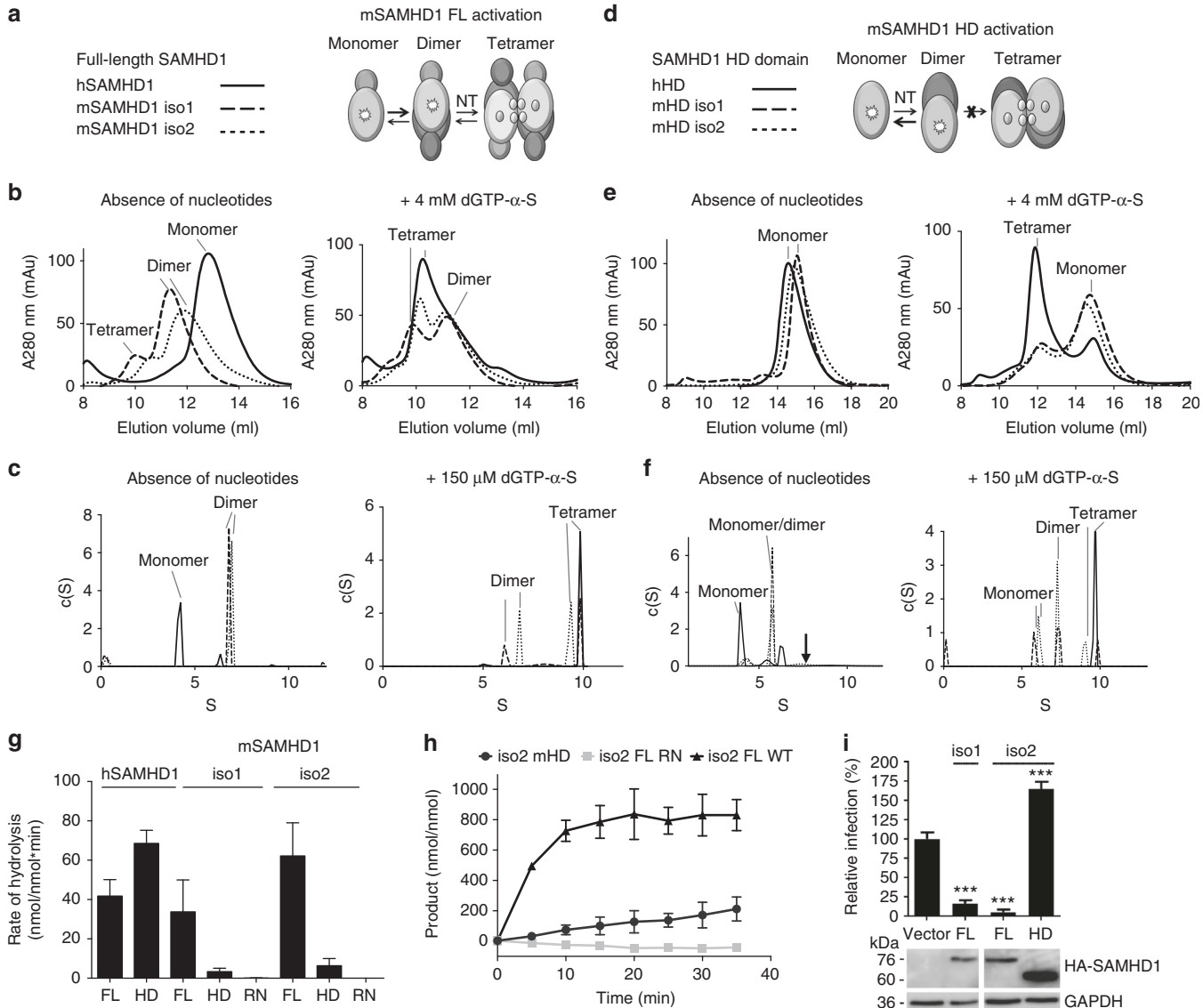

**Fig. 1** SAM domain of mSAMHD1 is required for tetramerization and function. **a**, **d** Cartoon schematics depicting oligomer states induced by adding nucleotides to full-length (FL) mSAMHD1 or its HD domain protein construct. **b**, **e** SEC analysis of purified SAMHD1 samples with (right) or without (left) nucleotides added. **c**, **f** AUC analysis of purified SAMHD1 samples with (right) or without (left) nucleotides added. The arrow in the left panel of (**f**) marks the position of HD dimer formed in presence of added nucleotides (right panel). **g** dNTPase assay for full-length, HD, and RN (inactive mutant) constructs of hSAMHD1 and mSAMHD1. Reaction products were quantified by HPLC. Each experiment was performed in triplicate. Error bars represent stand error (s.d.). **h** dNTPase activity assay for full-length, HD, and RN constructs of mSAMHD1. Reaction products were quantified by the malachite green assay. Each experiment was performed in triplicate. Error bars, s.d. (**i**) HIV-1 restriction by full-length mSAMHD1 iso1 and iso2, but not the HD domain of mSAMHD1 iso2. SAMHD1 proteins were stably expressed in U937 cells after PMA differentiation. HA-tagged SAMHD1 expression was confirmed by immunoblotting (uncropped images in Supplementary Fig. 7). GAPDH was used as a loading control. Single-cycle HIV-1 infection of vector control cells was set as 100%. Each experiment was performed in triplicate. Error bars represent standard deviation (s.d.) ***$p \leq 0.0001$

vector control cells (Fig. 1i). These data suggest that, unlike the HD domain of hSAMHD1 that has strong antiviral activities[26], the HD domain of mSAMHD1 is not sufficient for its anti-HIV function.

**Crystal structure of activated full-length mSAMHD1 tetramer.** We determined the crystal structures of the full-length mSAMHD1 protein, which furnishes our understanding of the fully functional state of the enzyme. The structure of full-length hSAMHD1 has remained elusive, despite extensive efforts by many groups; all published structures of hSAMHD1 only contain the catalytic HD domain[2,15,17]. Although a full-length hSAMHD1

protein was recently crystallized, the SAM domain could not be visualized, indicating inherent flexibility or disorder[29]. A structural understanding of the full-length SAMHD1 is important as the SAM domain is required for mSAMHD1 catalysis and is likely important for regulating hSAMHD1 activity.

The structure of the full-length mSAMHD1 iso1 protein with both allosteric sites occupied (GTP in Allo-site1 and dGTP in Allo-site2) was determined to a resolution of 3.5 Å, and hereafter is referred to as 2-Allo (Fig. 2a and Table 1). The 2-Allo structure was determined as a tetramer in I222 space group with one copy of the protein in each asymmetric unit of the crystal. The majority of the SAM and HD domains (residues 71–624) were built into the structure, although the C-terminal region including the

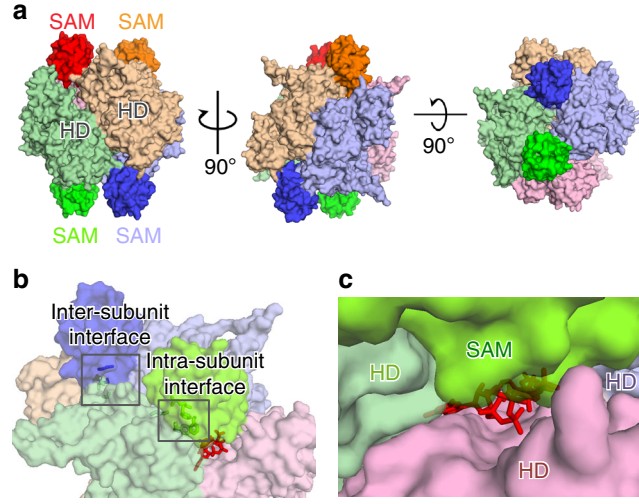

**Fig. 2** Crystal structure of full-length mSAMHD1 iso1 in complex with GTP and dGTP (2-Allo). **a** Surface representation of mSAMHD1 tetramer in three orthogonal views. Each subunit is shown in a different color (purple, orange, red, and green) with SAM domains highlighted. **b** Surface representation of the 2-Allo structure showing the allosteric binding pocket at the interface of three subunits (purple, red, and green). GTP and dGTP are shown with red sticks and the SAM-to-HD inter- and intra-subunit interactions are highlighted with selected interface residues shown in sticks. **c** Transparent surface representation of the SAM domain (lime green) capping the allosteric site

**Table 1 Data collection and refinement statistics (molecular replacement)**

|  | 2-Allo | 1-Allo | 0-Allo |
|---|---|---|---|
| *Data collection* |  |  |  |
| Space group | I 2 2 2 | P 21 21 2 | C 1 2 1 |
| *Cell dimensions* |  |  |  |
| $a, b, c$ (Å) | 85.68, 102.78, 145.80 | 112.01, 130.25, 90.90 | 164.66, 109.92, 162.12 |
| $\alpha, \beta, \gamma$ (°) | 90.0, 90.0, 90.0 | 90.0, 90.0, 90.0 | 90.0, 105.8, 90.0 |
| Resolution (Å) | 50–3.50 (3.56–3.50) | 50–3.40 (3.46–3.40) | 50–3.50 (3.56–3.50) |
| $R_{sym}$ or $R_{merge}$ | 0.15 (0.71) | 0.14 (1.00) | 0.15 (1.00) |
| Mean $I/\sigma I$ | 19.6 (2.0) | 10.7 (1.5) | 6.4 (1.4) |
| Completeness (%) | 98.0 (98.8) | 98.9 (99.9) | 98.8 (98.9) |
| Redundancy | 4.2 (4.3) | 3.6 (3.6) | 2.4 (2.3) |
| $CC^{1/2}$ | 0.54 | 0.43 | 0.53 |
| *Refinement* |  |  |  |
| Resolution (Å) | 20–3.50 (3.56–3.50) | 20–3.40 (3.46–3.40) | 20–3.50 (3.56–3.50) |
| No. of reflections | 8255 (796) | 18,591 (925) | 35,255 (1777) |
| $R_{work}/R_{free}$ | 24.2/29.3 (36.0/35.4) | 19.2/24.5 (28.8/37.3) | 24.2/27.1 (32.3/34.5) |
| *No. of atoms* |  |  |  |
| Protein | 4131 | 8271 | 16,514 |
| Ligand/ion | 94 | 64 | 4 |
| Water | NA | 20 | 63 |
| *B-factors* |  |  |  |
| Protein | 122 | 98 | 89 |
| Ligand/ion | 114 | 88 | 72 |
| Water | NA | 44 | 35 |
| *R.m.s. deviations* |  |  |  |
| Bond lengths (Å) | 0.013 | 0.008 | 0.009 |
| Bond angles (°) | 1.4 | 1.1 | 1.2 |

One crystal for 2-Allo, 1-Allo, and 0-Allo structures was used for data collection and structure determination. Values in parentheses are for highest-resolution shell

reported phosphorylation site (T634)[15,16,18,20] was not observed. Analysis of the 2-Allo structure in comparison to hHD structure reveals that the tetramer interface is fairly disordered with more than 25 residues unresolved (Supplementary Fig. 2a). This full-length SAMHD1 structure captures the relationship between the SAM and HD domains, and shows that each SAM domain bridges three HD domains at the tetramer interface (Fig. 2a, b).

The overall folds of both SAM and HD domains of mSAMHD1 are conserved with hSAMHD1. The mouse SAM (mSAM) domain aligns well with the existing NMR structure of the human SAM (hSAM) domain (PDBID 2E8O) and the crystal structure of the mandrill SAM domain[30] (PDBID 5AJA) with root-mean-square deviations (RMSD) of 0.77 Å and 1.00 Å, respectively (Supplementary Fig. 2b). The mHD tetramer has an RMSD of <0.6 Å when compared to hHD tetramer (PDBID 4BZB)[16] (Supplementary Fig. 2c). Although the overall architecture and fold of the individual HD domains are conserved between human and mouse, there are large differences in their relative orientations which affect the overall compactness of the tetramers (discussed in latter sections). The most significant deviation between mSAMHD1 and hSAMHD1 subunit structures occurs at the linker region between the mHD and mSAM domains, which is the last observed region of the hSAMHD1 structure crystallized in the full-length background[29] (black arrow, Supplementary Fig. 2d). This is consistent with the flexibility between the SAM and HD domains of hSAMHD1 and may explain why crystallization attempts have been unsuccessful with full-length hSAMHD1. Since mSAMHD1 and hSAMHD1 proteins share 72% sequence identity[18] and the overall folds of both SAM and HD domains are highly conserved, we believe our mSAMHD1 structure provides an excellent structural model for full-length hSAMHD1.

**Mouse SAM domain is essential for capping allosteric sites**. Our 2-Allo mSAMHD1 structure shows that the SAM domain contributes to the formation of the allosteric site pocket by interacting with the HD domains to stabilize the allosteric

nucleotides (Fig. 2c). Each allosteric site is formed as a deep cleft at the interface of HD domains from three subunits (Fig. 2b, c). Each SAM domain interacts intra- and inter-molecularly to contact the three HD domains, capping the nucleotides bound in the allosteric site pocket (Fig. 2c). These interactions explain the important role of the SAM domain in stabilizing the active tetramer (Fig. 1b, c) and hence the subsequent effects on mSAMHD1 enzymatic and antiviral activities (Fig. 1g, i). mSAMHD1's dependence on the SAM domain for allosteric activation is a striking feature that differentiates it from hSAMHD1. It is possible that the SAM domain has a different structural or regulatory role in the context of hSAMHD1.

**Importance of SAM-to-HD interactions for mSAMHD1 activity**. We further determined that the interactions between the SAM and HD domains are critical for mSAMHD1 enzymatic and antiviral activities, consistent with our biochemical and structural data that the SAM domain are important for both dimerization and tetramerization of the molecule. Located at the interface of three HD domains, the SAM domain engages in various interactions with HD domains from the same (intra-subunit) and adjacent subunits (inter-subunit) in the tetramer (Fig. 2b). These interactions help position the SAM domain in the correct orientation to cap the allosteric sites (Fig. 2b, c). The residues involved in these interactions are not conserved between the otherwise highly homologous mouse and human SAMHD1 proteins (Supplementary Fig. 3). The SAM domain is known to be dispensable for hSAMHD1 enzymatic and antiviral activities[2,12,26–28]. The functional integrity of hHD suggests that

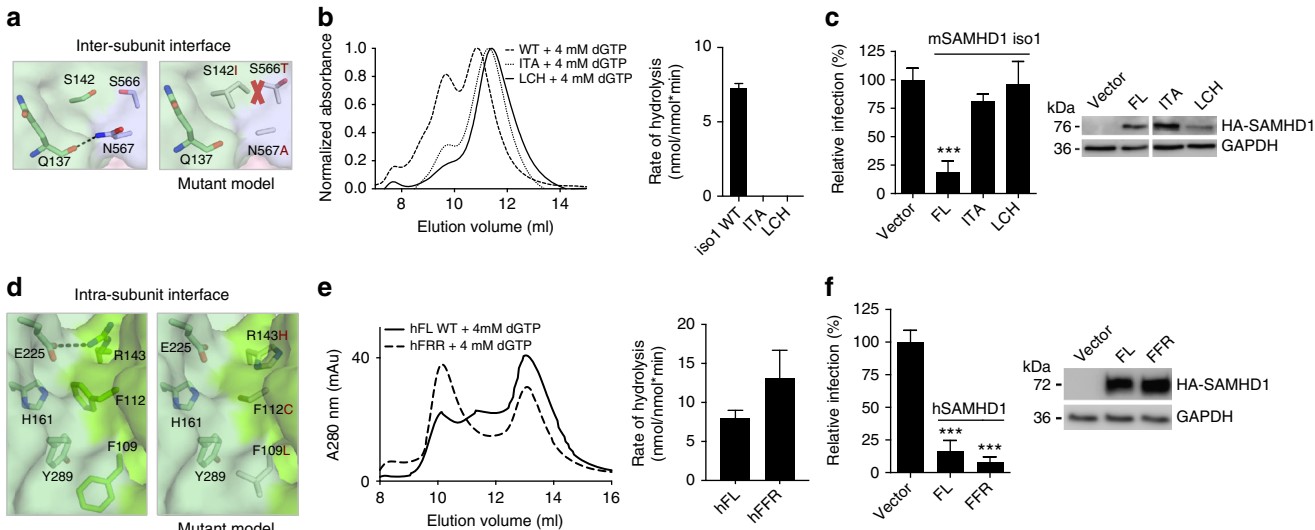

**Fig. 3** The importance of SAM-to-HD interactions for mSAMHD1 oligomerization and activities. **a**, **d** Left, transparent surface representation of SAM-to-HD interface with important residues shown in sticks. Right, model of SAM-to-HD interface after mSAMHD1 residues are mutated to the corresponding hSAMHD1 residues. The red cross in **a** indicates a steric clash. **b**, **e** Left, SEC analysis of mSAMHD1 (**b**) or hSAMHD1 (**e**) variants before and after incubation with dGTP-α-S. Right, dNTPase assay for wild type and mutant mSAMHD1 proteins (**b**) or hSAMHD1 (**e**). Products were quantified by the malachite green assay. Each experiment was performed in triplicate. Error bars, s.d. **c**, **f** HIV-1 restriction by full-length mSAMHD1 iso1 (**c**) or hSAMHD1 (**f**) (WT and mutants) stably expressed in U937 cells after PMA differentiation. Immunoblotting (uncropped images in Supplementary Fig. 7) confirmed HA-tagged mSAMHD or hSAMHD1 expression. GAPDH was used as a loading control. Single-cycle HIV-1 infection of vector control cells was set as 100%. Each experiment was performed in triplicate. Error bars, s.d. ***$p \leq 0.0001$

the critical SAM-to-HD interactions observed in mSAMHD1 are not required in hSAMHD1. This notion is supported by modeling the corresponding hSAMHD1 residues into the SAM-to-HD interfaces of mSAMHD1. This model predicts the disruption of the observed tetramer-stabilizing interactions (Fig. 3, Supplementary Figs. 3 and 4a).

To validate the importance of the observed SAM-to-HD interfaces, we first assessed the inter-subunit interaction's contribution to stabilizing the mSAMHD1 tetramer and maintaining its function. We tested the effect of mutating the key mSAMHD1 SAM-to-HD residues to their hSAMHD1 counterparts (Supplementary Fig. 3). At the inter-subunit interface, residues S566 and N567 of one HD domain interact across subunits with S142 and Q137 of the SAM domain from an adjacent mSAMHD1 molecule (Fig. 3a). Modeling a S142I/S566T/N567A triple mutant (ITA) onto our 2-Allo structure showed that S142I and S566T result in a steric clash and N567A disrupts a salt bridge with Q137 (Fig. 3a). As expected, our experimental results showed that the ITA mutant was defective in forming a tetramer and displayed a substantial reduction in dNTPase activity (Fig. 3b). Similarly, this ITA mutant did not confer HIV-1 restriction compared to full-length wild-type mSAMHD1 (Fig. 3c), demonstrating that these SAM-HD inter-subunit interactions are important for mSAMHD1 function in vivo.

Next, we assessed the importance of the intra-subunit SAM-to-HD interactions in oligomerization and enzymatic function (Fig. 2b). At the mSAMHD1 intra-subunit interface, SAM domain residue R143 forms a salt bridge with HD domain residue E225 and the aromatic residues F109 and F112 from the SAM domain form stacking interactions with H161 and Y289 from the HD domain, respectively (Fig. 3d). We swapped these residues with their human counterparts to make the triple mutant F109L/F112C/R143H (LCH) (Supplementary Fig. 3). Modeling of these mutations shows that they disrupt the electrostatic and stacking interactions that position the SAM domain to stabilize the tetramer (Fig. 3d). Indeed, the LCH mutation completely abolished mSAMHD1 tetramer formation and dNTPase activity

(Fig. 3b). This mutation also significantly reduced the HIV-1 restriction function of mSAMHD1, albeit the mutant expressed lower than wild type (WT) full-length protein (Fig. 3c).

In a complimentary study, we created the L77F/C80F/H111R (FFR) mutations in hSAMHD1 (Supplementary Fig. 4a) to introduce the SAM-to-HD interactions observed in mSAMHD1. Because these interactions are important for stabilizing the mSAMHD1 tetramer, we expected their introduction into hSAMHD1 to have a similar tetramer-stabilizing effect. Indeed, our results show that this triple mutant displayed higher dNTPase activity and formed a more stable tetramer than WT hSAMHD1 (Fig. 3e). Correspondingly, we observed a twofold increase in HIV-1 restriction by the FFR mutant compared to full-length WT hSAMHD1 (Fig. 3f). These data demonstrate that enhancing interactions between the SAM and HD domains of hSAMHD1 lead to a more stable tetramer with higher enzymatic and HIV-1 restriction activities.

**mSAMHD1 undergoes a multi-step activation process.** In addition to the 2-Allo structure, we also captured crystal structures of full-length mSAMHD1 in two more activation intermediate states: no allosteric nucleotides bound (No-Allo) and one allosteric nucleotide bound to Allo-site 1 only (1-Allo) (Fig. 4a). The No-Allo crystal was obtained without the addition of nucleotides in the crystallization mixture and diffracted to 3.5 Å resolution. The 1-Allo crystal was crystallized with dGTP-α-S in the crystallization mixture and diffracted to 3.4 Å resolution. The No-Allo and 1-Allo mSAMHD1 structures also crystallized as tetramers in different space groups (C2 and P2₁2₁2) with 4 or 2 copies of mSAMHD1 in the asymmetric unit, respectively.

To confirm that these intermediate states, which lack the necessary nucleotides at Allo-site 2, are inactive forms of the enzyme, we tested whether the residues that interact with the Allo-site 2 nucleotides are critical for catalysis (Supplementary Fig. 4b, c). These residues at the Allo-site 2 pocket do not interact with the Allo-site 1 nucleotides nor directly contribute to

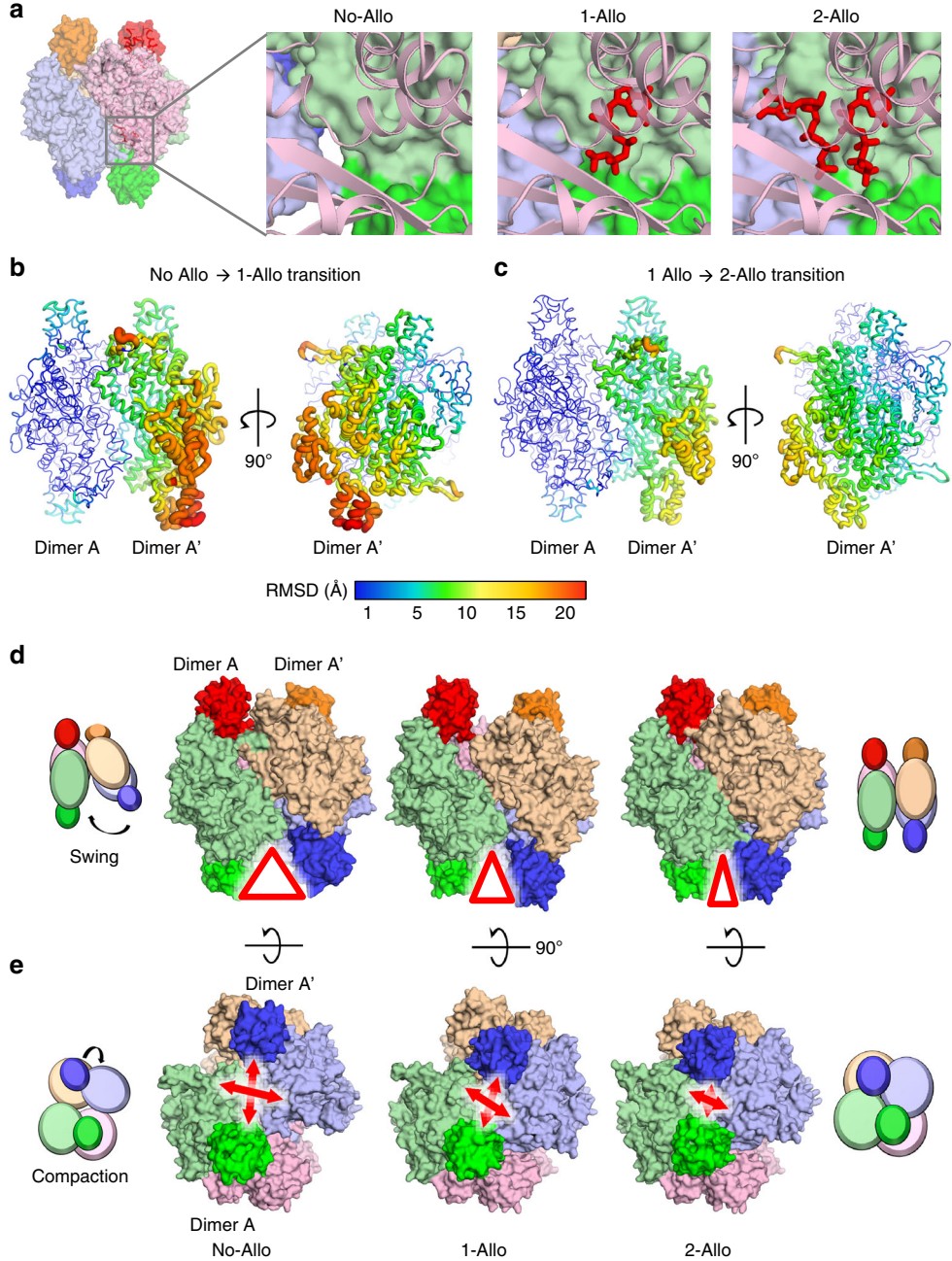

**Fig. 4** Crystal structures of mSAMHD1 activation intermediates. **a** Surface and ribbon representations of the mSAMHD1 structures at the allosteric site. Nucleotides are shown in red sticks and each subunit is colored uniquely. Left, No-Allo structure with no nucleotides bound. Middle, 1-Allo structure with dGTP bound to Allo-site 1. Right, 2-Allo structure with GTP and dGTP bound to Allo-site 1 and Allo-site 2, respectively. **b**, **c** Putty representation comparing the No-Allo and 1-Allo mSAMHD1 structures (**b**) and comparing the 1-Allo and 2-Allo structures (**c**) in two orthogonal views. The structures are aligned by superposition of dimer A. RMSD values are represented using the color spectrum and the thickness of the coil. **d**, **e** Surface representation of the No-Allo (left), 1-Allo (middle), and 2-Allo (right) structures showing conformational differences. Red triangles and arrows highlight the separation between a pair of SAM domains (green and purple). Cartoon schematics of the No-Allo (left) and 2-Allo structures (right) indicate the swinging (top row) and compacting (bottom row) motions of dimer A' toward dimer A. Double-headed arrows highlight the distances between SAM domains as the tetramer becomes more compact

tetramerization. Mutating these residues in both isoforms of mSAMHD1 abolished dNTPase activity (Supplementary Fig. 4c). These results demonstrate the importance of nucleotide-recognition residues, and hence nucleotide binding at Allo-site 2, for catalytic activity. Although mSAMHD1 is capable of forming tetramers without nucleotides bound to Allo-site 2, these states are inactive and likely represent activation intermediates of the enzyme.

To understand the differences between the three tetramer states, we systemically analyzed the conformations of the mSAMHD1 subunits that assemble the tetramer (Supplementary Fig. 5). First, we aligned the subunits within each structure and between the three states and found that their RMSDs are very low (0.5–1.0 Å) (Supplementary Fig. 5a–c). This indicates that the overall conformations of the individual monomeric subunits that make up the tetramer do not change much between the states.

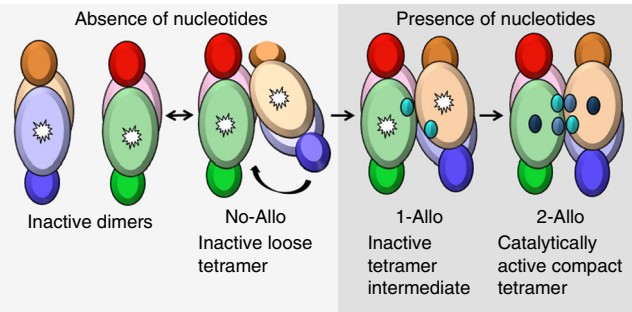

**Fig. 5** Model of mSAMHD1 allosteric activation. In the absence of nucleotides, mSAMHD1 primarily exists as inactive dimers that can associate to form loose inactive tetramers. After the first allosteric nucleotide binds, the rigid dimer components rearrange to form a more compact but inactive intermediate. Upon binding of the second allosteric nucleotide, mSAMHD1 rearranges further to form a catalytically active, compact tetramer

Both the No-Allo and 2-Allo tetramers are symmetrical and obey the *222* point-group symmetry with three perpendicular twofold axes. However, this symmetry is broken in the 1-Allo tetramer, which is a dimer of dimers, with only one overall twofold axis. Interestingly, the dimeric building block (designated as dimer A) remains rigid with minor conformational variations (RMSDs 0.7–1.3 Å) among the three different tetramer states (Fig. 4b, c, and Supplementary Fig. 5d, e). This suggests that the dimer may serve as an intermediate assembly unit for the formation of the active mSAMHD1 tetramer.

Despite relatively rigid monomer and dimer building blocks, the three mSAMHD1 tetramers (No-Allo, 1-Allo, and 2-Allo) (Fig. 4a) display large conformational differences (RMSDs 2.4–3 Å) with varying compactness (Fig. 4d, e and Supplementary Fig. 5f). Aligning the three structures by the rigid dimer A highlights the relative movement of the other half of the tetramer (designated as dimer A′, Fig. 4d). Starting from the most open No-Allo state, dimer A′ swings progressively toward dimer A via the 1-Allo-state to close the tetramer interface in the 2-Allo state (Fig. 4d and Supplementary Movie 1). The consequence of the swinging motions is a formation of a compact tetramer highlighted by the distance between the SAM domains of dimers A and A′ (Fig. 4e and Supplementary Movie 2). These significant rearrangements between the two dimers occur as nucleotides sequentially bind the allosteric sites to activate the enzyme.

## Discussion

SAMHD1 plays an important role in controlling cellular dNTP pool levels and has been identified as an HIV-1 restriction factor in non-dividing cells[1,3,31]. Additionally, mutations in SAMHD1 are associated with autoimmune disease states, such as AGS[8]. In this study, we provide a rigorous biochemical analysis and crystal structures of full-length mouse SAMHD1 that capture the interactions between the SAM and HD domains. This study establishes a mechanistic framework for understanding the key differences between human and mouse SAMHD1 activities, allosteric activation and retroviral restriction properties of mSAMHD1.

Our results show that a key difference between the human and mouse enzyme is that mSAMHD1, but not hSAMHD1, requires the SAM domain for dNTPase and antiviral activities. This is likely due to the SAM domain's contribution to nucleotide binding at the allosteric site. By capping the allosteric nucleotides in the pocket, the SAM domain is essential to the formation of active mSAMHD1 tetramers. Our analysis shows that SAM-to-HD interactions are crucial for mSAMHD1 oligomerization and function. Mutating key

residues involved in SAM-to-HD interactions substantially decreased the ability of mSAMHD1 to form a tetramer, which results in significantly reduced dNTPase and antiviral activities. While mSAMHD1 depends on its SAM domain to stabilize the active tetramer, residues involved in these interactions are not conserved in hSAMHD1. Consistent with their tetramer-enhancing role, we were able to engineer these SAM-to-HD interactions into hSAMHD1 to create a more stable tetramer with higher dNTPase and antiviral activities. This hyper-stable tetramer hSAMHD1 construct may be useful for future biochemical and structural studies that will provide insights into the effect of the SAM domain in regulating the activities of hSAMHD1.

The SAM domains of hSAMHD1 and mSAMHD1 may play different roles in the regulation of enzymatic and other cellular functions. In contrast to mSAMHD1, hSAMHD1 does not require the SAM domain for stable formation of the tetrameric catalytic core. In fact, the hHD domain is more active than the full-length protein, so it is possible that the SAM domain imposes a negative regulatory effect on the catalytic activity of hSAMHD1. These opposing effects of SAM domain on mSAMHD1 and hSAMHD1 activities might reflect differences in the sensitivity to nucleotides. It has been shown that the nucleotide levels in human cells are lower than those in mouse cells[32]. As a result of being exposed to higher concentrations of nucleotides, mHD may have evolved lower affinities for nucleotide activators whose binding are enhanced by the SAM domain. Conversely, the hHD itself may have developed a higher affinity for nucleotides at relatively lower nucleotides concentrations. It also remains possible that the SAM domain plays other important roles in regulating these enzymes in the cell or processes yet undiscovered functions.

The existence of mSAMHD1 activation intermediates may also explain the previous biochemical observations that mSAMHD1 is active in the absence of additional nucleotide activators[33]. Bloch et al.[33] showed that mSAMHD1 tetramer has dNTPase activity in the absence of additional dGTP and suggested that this activity is from the nucleotides bound to SAMHD1 during purification. We also observed dimer and tetramer species during our purification of mSAMHD1, thus we tested whether these states of mSAMHD1 exhibit dNTPase activity in the absence of added GTP/dGTP. We carried out a dNTPase activity assay in the presence of dATP, which cannot bind to Allo-site 1 to assemble an active tetramer. Thus, activity would only be detected if mSAMHD1 is purified with Allo-site 1 GTP or dGTP bound. We found that the dimer of mSAMHD1 did not exhibit hydrolysis activity similarly to the inactive RN control (Supplementary Fig. 1c), indicating that mSAMHD1 assembles dimers in the absence of any bound nucleotides. In contrast, the purified tetramer did hydrolyze dATP, albeit at a very slow rate, requiring incubation times up to 2 h (Supplementary Fig. 1c). Since our experiment did not include GTP or dGTP, which are the only allosteric activators to bind Allo-site 1[15–17], we conclude that some populations of mSAMHD1 protein were purified with GTP or dGTP bound to the allosteric site 1, consistent with the previous observations[33]. These tetramers likely represent activation intermediates similar to our 1-Allo structure, and they are not fully active states of the enzyme.

Our study indicates that the allosteric activation of mSAMHD1 and hSAMHD1 is considerably different. While hSAMHD1 transitions from a monomer or dimer to a tetramer with the addition of allosteric nucleotides, mSAMHD1 seems to have a more complex landscape of oligomerization states. In the absence of nucleotides, mSAMHD1 exists in an equilibrium between stable dimers and inactive tetramers. Three tetramer states exist with 0 (No-Allo), 1 (1-Allo), or 2 (2-Allo) nucleotide(s) bound at each allosteric site. Our No-Allo and 1-Allo structures show that these inactive tetramer intermediates are composed of two relatively rigid dimers with loose tetramer interfaces. In the presence

of similar concentrations of nucleotides, mSAMHD1 forms fewer tetramers in solution than hSAMHD1. Our 2-Allo mSAMHD1 structure shows that even this activated form of mSAMHD1 is still less compact than hSAMHD1, indicating an overall less stable tetramer.

A model of the mSAMHD1 activation process emerges from our structural and biochemical results, which includes a series of inactive tetramer intermediates with increasing numbers of allosteric nucleotides bound (Fig. 5). Initially, inactive mSAMHD1 exists in solution predominantly as dimers that can associate into loose tetramers. Nucleotide binding to Allo-site 1 causes the dimer components to swing around a pivot point and close in toward each other. Subsequent nucleotide binding to Allo-site 2 brings the dimers even closer together and positions the SAM domains to cap the allosteric sites for stable binding of the allosteric nucleotides. This activation model differs from the one proposed for hSAMHD1, as there is no evidence to suggest that hSAMHD1 can form tetramers in the absence of nucleotides. In the case of hSAMHD1, nucleotides are required for oligomerization whereas for mSAMHD1, nucleotides are required for conformational change and activation of the preformed tetramers.

Although mSAMHD1 appears to form less tetramers than hSAMHD1 at the same nucleotide concentration, their dNTPase activities are similar, indicating a higher activity for activated mSAMHD1. In fact, it has been previously shown that hSAMHD1 has a lower $K_{cat}$ than that of mSAMHD1[34]. This could be advantageous as mouse cells have a higher concentration of nucleotides than that of human cells[1,35]. Thus, mSAMHD1's relative insensitivity to nucleotides, and its slow and complex activation, ensure that mSAMHD1 is not continuously hydrolyzing dNTPs. These differences in oligomerization and enzymatic activity may help to explain the differences between human and mouse SAMHD1 and inform the improvement of a mouse model for the study of HIV-1, certain cancers and autoimmune diseases.

## Methods

**Protein expression and purification**. N-terminal 6× His-tagged mSAMHD1 constructs (Supplementary Table 1) were expressed in *Escherichia coli* BL21 (DE3). Cells were resuspended in 50 mM Tris, pH 8, 500 mM NaCl, 20 mM imidazole, 5 mM MgCl$_2$, 0.5 mM TCEP) and lysed using a microfluidizer. Cell debris was clarified by centrifugation at 26,892×*g* for 25 min. The mSAMHD1 constructs were purified using Ni-NTA affinity and size-exclusion chromatography in a final buffer of 50 mM Tris-HCl, pH 8.0, 150 mM NaCl, 5 mM MgCl$_2$, and 0.5 mM TCEP (SAMHD1 buffer).

**Analytical size-exclusion chromatography**. Purified samples of mSAMHD1 protein (2 mg/ml, 200 μl) mixed with dGTP or GTP (4 mM final concentration) were applied to a Superdex 200 10/300 GL column (GE Healthcare) pre-equilibrated in SAMHD1 buffer. The UV absorbance at 280 nm was recorded to monitor the elution of SAMHD1 oligomers.

**Analytical ultracentrifugation**. SAMHD1 constructs was prepared at 0.8–1.3 mg mL$^{-1}$ concentration in the SAMHD1 buffer and mixed with a final concentration of 150 μM nucleotides. Sedimentation velocity experiments were performed with the protein alone and in the presence of nucleotides samples with a Beckman XL-I analytical ultracentrifuge at 169,167×*g* at 20 °C with an An60-Ti rotor. SEDNTERP (http://sednterp.unh.edu/) was used to calculate the sample partial specific volume, buffer density and viscosity. SEDFIT[36] was used to analyze the velocity data.

**Crystallization and data collection**. mSAMHD1 protein in SAMHD1 buffer was mixed with various combinations of nucleotides at 0–7 mM concentrations (Table 1) and incubated at 4 °C for 15 min before crystallization. The apo WT structure was crystallized in the absence of nucleotides, the 1-Allo WT structure was co-crystallized in the presence of 4 mM dGTP-α-S, with the crystals optimized after rounds of streak seeding. The 2-Allo RN structure was co-crystallized with 1 mM GTP and 6 mM dGTP. All crystallization was done using the microbatch under-oil method. Protein sample at 5 mg/mL was mixed with crystallization buffer (100 mM SPG (Qiagen) buffer, pH 7.4, 25% PEG 1500) at a 1:1 ratio (1 μL protein:1 μL precipitant) and incubated at 25 °C. Crystals were cryoprotected with a mixture of crystallization buffer and 25% (Vol/Vol) glycerol before freezing in liquid nitrogen. All diffraction data were collected at the Advanced Photon Source

at 100 K temperature. The 2-Allo data set was collected at the beamline 24-ID C with wavelength at 0.91640 Å. 1-Allo and 0-Allo data sets were collected at the beamline 24-ID E with wavelength 0.97920 Å and 0.97918 Å, respectively. Data statistics are summarized in Table 1.

**Structure determination and refinement**. The structures were solved by molecular replacement using PHASER[37]. The search models included previously published SAMHD1 tetramer structure (PDBID 4BZB), with all bound nucleotides removed, and the SAM domain from mandrill SAMHD1 (PDBID 5AJA). The models were refined with iterative rounds of TLS and restrained refinement using Refmac5[38] followed by rebuilding the model to the $2F_o–F_c$ and the $F_o–F_c$ electron density maps (stereo images of density map in Supplementary Fig. 6) using Coot[39]. The electron density maps were improved by multi-domain (SAM and HD domains) multi-crystal averaging using DMMulti[40]. Applying the improved phases as restraints further improved the refinements of the No-Allo and 1-Allo structures. Refinement statistics are summarized in Table 1. The Φ–Ψ angles of 95.7% of residues of the 2-Allo structure, 95.3% of 1-Allo structure and 95.2% of No-Allo structure lie within favored regions of Ramachandran plot, and those of all residues lie in the allowed regions.

**Malachite green colorimetric assay of SAMHD1 activity**. All assays were performed at 25 °C in SAMHD1 buffer. Each 40 μL reaction, containing 10 μM pyrophosphatase, 0.5 μM SAMHD1, and 125 μM substrate or allosteric activator was quenched with 40 μL 20 mM EDTA after 15 min. Subsequently, 20 μL Malachite Green reagent was added to the solution and developed for 15 min before the absorbance at 650 nm was measured[41].

**HPLC assay of SAMHD1 activity**. Nucleotide hydrolysis reactions were carried out in the SAMHD1 buffer containing 0.5 mM GTP in a 500 μL reaction volume, with various concentrations of dNTPs. Reactions were initiated by the addition of SAMHD1 at a final concentration of 500 nM to the dNTPs solution and incubated in a 37 °C water bath. Reactions were terminated with a 5× dilution into ice-cold buffer containing 10 mM EDTA at various time points. Samples were deproteinized by spinning through an Amicon Ultra 0.5-ml 10-kDa filter (Millipore) for 20 min at 16,000×*g*. Samples were analyzed by HPLC with a Synergi C18 column 150 × 4.6 mm (Phenomenex). The column was pre-equilibrated in 20 mM ammonium acetate, pH 4.5 (buffer A) and samples were eluted at a flow rate of 1 ml/min with a gradient of methanol (buffer B) over 14 min. UV absorption was recorded at 260 nm.

**HIV-1 restriction assay**. U937 Cells (ATCC CRL-1593.2) were obtained from the American Type Culture Collection (ATCC) and cultured with RPMI medium with 10% FBS. Stable U937 cells expressing full-length or mutant SAMHD1 were generated by spinoculation with concentrated lentiviral vectors and selection by puromycin (0.8 μg/mL)[42]. U937 cells were differentiated using phorbol 12-myristate 13-acetate (PMA) (100 ng/mL) for 24 h, PMA was removed and cells were cultured for a further 24 h. All cell lines were tested negative for mycoplasma contamination using a universal mycoplasma detection kit (ATCC, #30-101-2 K). To quantitate HIV-1 infection in PMA-differentiated SAMHD1 expressing stable U937 cell lines, cells were infected with single cycle, vesicular stomatitis virus protein G (VSV-G) pseudotyped luciferase reporter HIV-1 (HIV-Luc/VSV-G) at a multiplicity of infection of 0.5 for 2 h before removal of virus. Cells were lysed in 1× reporter lysis buffer (Promega) and analyzed by luciferase assay using a commercially available kit (Promega)[42].

**Statistical information**. The average ± standard errors and standard deviations were calculated from multiple separate experiments as indicated in each figure legends and the results are shown in each graph. Error bars in HIV restriction data show standard deviation of at least three independent experiments as analyzed by one-way ANOVA with Dunnett's multiple comparison test using GraphPad Prism 5.0.

**Data availability**. Coordinates and structural factors have been deposited in the Protein Data Bank under accession codes 6BRK for 2-Allo, 6BRH for 1-Allo and 6BRG for No-Allo structures. Other data are available from the corresponding authors upon reasonable request.

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

## Acknowledgements

We thank B. Slater, J. Wang, T.C. Cheng, S.S. Smaga, T. Cheng, and C.S. Gelais for technical assistance, and K. Digianantonio and W. Wang for assistance and discussion. We also thank the staff at the Advanced Photon Source beamline 24-ID C and E. This work was supported in part by the NIH grants AI102778 (Y.X.), AI120845 (X.J.), AI104483 (L.W.). O.B. was supported by the predoctoral program in Cellular and Molecular Biology T32 GM007223 and by the National Science Foundation Graduate Research Fellowship. K.M.K. was supported by the NIH T32 grant GM008283. J.M.A. was supported by C. Glenn Barber funds from the College of Veterinary Medicine at The Ohio State University.

## Author contributions

O.B. and C.T. contributed equally to this work. C.T., O.B., K.M.K., performed protein expression, purification. C.T. and X.J. cloned, and performed crystallization, data collection, and structure determination. O.B. performed structural analysis. C.T. and O.B. performed SEC assays. O.B. and K.M.K. performed AUC assays. O.B. and K.M.K. performed activity assays. J.M.A. performed cell-based experiments. O.B., C.T., K.M.K., J.M.A., L.W., X.J. and Y.X. designed the experiments. C.T., O.B., K.M.K., J.M.A., L.W., X.J. and Y.X. analyzed the data. O.B., K.M.K., X.J., L.W. and Y.X. wrote the manuscript.

## Additional information

**Competing interests:** The authors declare no competing financial interest.

