## [Peer Review File · Nature Communications]

REVIEWERS' COMMENTS:

Reviewer #1 (Remarks to the Author):

Buzovetsky et al. The SAM domain of mouse SAMHD1 is critical for its activation and regulation.

The authors have determined the crystal structures of the full length mSAMHD1 protein in three different activation states. They have determined that mSAMHD1 has a more complex nucleotide induced regulatory process than the human SAMHD1 protein. The authors show that mSAMHD1 requires the SAM domain for catalysis and for antiviral activity, and their structures reveal the protein interactions required for tetramerization and formation of the allosteric binding sites. Importantly, their structures of 0,1, & 2 nucleotides in the regulatory sites, reveal allosteric intermediates that indicate SAMHD1 undergoes a multi-step activation process.

The manuscript is generally well written and the figures are clear. This work is an important new finding for the SAMHD1 protein and will be of interest to a broad group of researchers. The authors may want to address the points below.

1. Table 1. The 3 structures are determined to a resolution of about 3.4-3.5 Ang. However, the highest resolution shell for each structure has a Rmerge of 0.71 or 1.0, which seems very high. Given the low signal to noise in those resolution shells the authors should provide some other justification, such as $CC1/2$, for choosing this data cutoff limit.
2. $I/\sigma I$ should be mean $I/\sigma I$

Reviewer #2 (Remarks to the Author):

The manuscript by Buzovetsky et al describes the first crystal structure of a mammalian SAMHD1 enzyme (mouse) that contains an observable N-terminal SAM-domain. In all previous structures of the human enzyme, this domain is disordered and not observable. The authors find that in the mouse enzyme the SAM domain docks against the catalytic HD domain and plays a role in stabilizing the tetramer and in regulating access of nucleotides to activator sites. None of these behaviors match that of the human enzyme and indicate a surprising change in the regulation mechanism of the mouse version. The paper is well-written and the structures and associated biochemical data provide useful information about the role of the SAM domain in the mouse system. My only disappointment with the work is that we still have no idea what role the SAM domain of the human enzyme plays in its many biological functions. This lessens my enthusiasm for this otherwise solid work.

We thank the reviewers for their comments and highlighting our manuscript's impact on SAMHD1 studies as well as the broad field. We have addressed all of the reviewers' comments as detailed below.

Reviewer #1

1. The reviewer noted that the highest resolution shell for each structure has a high Rmerge and given our signal to noise ratio, the reviewer had asked for CC1/2 values to justify this data cutoff limit.

We added the CC1/2 values to Table 1.

2. The reviewer pointed out that $I/\sigma I$ should actually be referred to as mean $I/\sigma I$ in Table 1.

We have made this correction to Table 1.

Reviewer #2

The reviewer thought that our manuscript is well written and that the experiments provide useful information about the role of SAM domain in mouse system. He/she did not have any further questions to be addressed.